# Stratospheric Variability at a glance - Analysis of the intra decadal timescale and the QBO

Duy Cai<sup>1</sup>, Martin Dameris<sup>1</sup>, Hella Garny<sup>1</sup>, Felix Bunzel<sup>2</sup>, Patrick Jöckel<sup>1</sup>, and Phoebe Graf<sup>1</sup>

 <sup>1</sup>Deutsches Zentrum für Luft- und Raumfahrt, Institut für Physik der Atmosphäre, Oberpfaffenhofen, Germany
 <sup>2</sup>Max-Planck-Institut für Meteorologie, Hamburg, Germany
 *Correspondence to:* D.S. Cai (duy.cai@dlr.de)

**Abstract.** In this study the stratospheric variability is analysed from decadal to seasonal timescales. Relevant processes for the decadal timescale are identified by means of power spectral analysis. The inspection of the ERA-Interim reanalysis data set shows considerably high variability at the 12 and 6 months period. But also in the extra tropical region at intra-annual to seasonal timescales

- 5 clear peaks in the power spectrum arise. In addition to that, the quasi-biennial oscillation (QBO) obviously contributes to the stratospheric variability at decadal timescales. Regarding the power spectrum of EMAC 2.52 model simulations, only a model configuration with a vertical resolution smaller than 1 km in the stratosphere is capable to capture the relevant features of the spectrum. In particular, the model with a coarser distribution of vertical levels cannot reproduce the QBO signal.
- 10 The analysis of the corresponding wave spectra reveals that, if the vertical resolution is insufficient, primarily the Mixed-Rossby-Gravity waves cannot be adequately reproduced. Estimates made by linear wave theory show that for reanalysis data the Mixed-Rossby-Gravity waves with equivalent depths between 50 m to 250 m are relevant for the QBO. In order to resolve these relevant waves, model simulations need to consider a vertical resolution of at least 1 km.

# 15 1 Introduction

Reliable information on weather and climate are of increasing interest for economy, politics and society. In particular decadal timescales become more and more important. This is also reflected by governmental efforts to support the development of decadal numerical prediction systems, e.g. the German research project for decadal predictions (Mittelfristige Klimaprognosen, MiKlip; Pohlmann et al., 2013).

This study focuses on stratospheric processes relevant for the dynamical variability on timescales ranging from months to decades. In recent years several studies have proven the relevance of the stratosphere for the tropospheric behaviour. In particular, dynamical links, such as stratospheric anomalies propagating downward into the troposphere and even reaching the ground, are docu-

- mented for observational as well as numerical model data (e.g. Baldwin and Dunkerton, 2001; Slingo, 2013; Runde et al., 2016). Kidston et al. (2015) showed for monthly, seasonal, decadal and even to centennial timescales, profound stratospheric implications on the troposphere can be detected. In particular, seasonal forecasts covering stratospheric processes showed promising signs to improve the representation of the troposphere (e.g. Baldwin et al., 2003; Charron et al., 2012; Sigmond et al.,
- 2013). Actually, some mid-range tropospheric forecast skill appears to result from stratospheric processes (e.g. Mukougawa et al., 2009; Scaife et al., 2014a; Seviour et al., 2014). Also it is shown that increasing the stratospheric resolution of the model has the potential to improve the extended range (couple of weeks) forecast skill (Roff et al., 2011).
- However, for the decadal timescale it remains to be proven that climate prediction benefits from 35 the precise representation of stratospheric processes (Smith et al., 2012; Kidston et al., 2015). A prominent example of these processes are the interannual fluctuations in the tropical stratosphere with zonal wind anomalies persisting for many months describing the quasi-biennial oscillation (QBO). Due to this interannual memory of the tropical stratosphere (Scott and Haynes, 1998), the QBO could be a considerable source of climate predictability on seasonal to decadal timescales.
- Scaife et al. (2014b) demonstrate skillful predictability of the QBO in decadal forecast systems. However, there is room for improvements regarding the predictability of the QBO and in particular the QBO winter surface teleconnections are not clearly captured.

The open questions and tasks, which have to be tackled now are the identification of the relevant stratospheric processes on the decadal timescale, and the analysis of their representation in climate models. This paper is organised as follows. Section 2 gives a brief overview of the model and reanalysis data sets used as well a short description of the power spectral analysis. Section 3 presents the power spectral analysis at a decadal timescale for the global wind field. In section 4 the tropical variability, particularly the QBO is investigated in more detail. The final section 5 gives a discussion and the conclusions.

#### 50 2 Data and Methods

#### 2.1 Model and Reanalysis data

In our study, we analyse simulations performed with the ECHAM/MESSy Atmospheric Chemistry (EMAC) model version 2.52 (Jöckel et al., 2010) and the reanalysis data ERA-Interim (ERAI). EMAC is a numerical chemistry and climate model system describing physical and chemical pro-

55 cesses from the troposphere to the middle atmosphere. In the present simulations it uses the Modular Earth Submodel System 2 (MESSy2), which is a software package providing the infrastructure of generalised interfaces for the standardised control and coupling of the so called submodels (i.e. submodels encompass low-level earth-system-model components e.g. dynamic cores, physical pa-

rameterisations, chemistry packages, diagnostics). The underlying core atmospheric model is the 5th 60 generation European Centre Hamburg general circulation model (ECHAM5; Roeckner et al., 2003).

In this study we perform two so-called time slice simulations. Time-slice simulations describe a specific climate equilibrium state. Technically the boundary conditions are repeated with each year and therefore only a seasonal variability is allowed. The simulations performed here are all conducted with a prescribed annual cycle of the greenhouse gas concentrations (i.e.  $CO_2$ ,  $N_20$ ,  $CH_4$ )

- of the year 2000 derived from values recorded in the IPCC (2007) and a zonally symmetric ozone climatology based on the work of Paul et al. (1998). Likewise, the model top at 80 km remains the same in each simulation which also means that the middle atmosphere dynamics is included. The reference simulation (BASE) is the realisation of a 20 year time-slice experiment, with additional 10 years spin-up considered as the climate equilibrium state around the year 2000. The horizontal
- resolution is T42 corresponding to an approximately 2.8°x 2.8°quadratic Gaussian grid. The vertical extent from the surface to the middle atmosphere with an upper model lid centred at 0.01 hPa is distributed over 90 vertical levels. With this model configuration vertical features finer than 1 km are resolvable at stratospheric altitudes. The sea surface temperatures (SSTs) are prescribed and utilise the annual cycle of the 10 year climatological mean (from 1995 to 2004) of the Atmospheric
- Model Intercomparison Project (AMIP) SSTs described by Hurrell et al. (2008). The second simulation (hereinafter LR) uses a lower vertical resolution with 47 vertical layers, which corresponds approximately to a 2 km vertical resolution at stratospheric altitudes.

Apart from the vertical resolution, the simulation setup of LR is identical to the simulation setup described for the BASE simulation (detailed information about technical settings and differences of vertical model setups have been documented by Jöckel et al. (2016)).

In this study, ERAI (Dee et al., 2011) is considered as a reference for the evaluation of the model results. ERAI is the third generation reanalysis product of the European Centre for Medium-Range Weather Forecast (ECMWF). This data set is provided at a spatial resolution of 0.75°x 0.75° and 60 vertical levels up to 0.1 hPa top, covering several decades from 1979 to today. For the purpose of

85 comparison the horizontal resolution of ERAI is regridded to T42, i.e. the same as used in the model simulations. Furthermore, we choose the 21 year period (1990–2010) and remove the least squares linear trend of this transient time series in order to obtain a similar climate state as simulated in the model. The analysed variables are temperature and zonal wind.

# 2.2 Methodology

The method used for the first part of this study is based on the power spectral analysis following the considerations of Hayashi (1971). Since we are interested in intra-decadal timescales the spectral quantities were calculated from several 120-months segments of a multidecadal stratospheric zonal wind dataset at 30 hPa, with a temporal resolution of monthly means. The individual segments are realised by a 120-month time window, which is shifted successively by 6 months. The results are

- not sensitive to the choice of this shift (tests were also conducted with a 3 months and a 6 months shift) and also different stratospheric levels show qualitatively similar results (not shown). To ensure comparability, the linear trend of transient timeseries is removed by a least square fit. Then, for each 120-months segment a 10% split-cosine-bell tapering is applied in order to minimise the effects of spectral leakage (e.g., von Storch and Zwiers, 2001; Bloomfield, 2004). After tapering, complex
- FFTs in time and longitude are performed. In our analysis the variance is given by the multiplication of the complex number with its corresponding complex conjugated number. In the present analysis we sum over all wavelengths and average over all available segments of the multidecadal dataset. Thus, the resulting variance only depends on latitude and frequency. For statistical considerations the number of degrees of freedom (dof) is needed. In our example of ERAI the total number of dof
- is about 6 (3 independent realisations x 2 temporal and spatial). In the following this method will be shortly named "decadal power spectrum".

To analyse equatorial waves on typical seasonal timescales, the wavenumber-frequency analysis presented by Wheeler and Kiladis (1999) is utilised. In their work they made use of the spatial characteristics of equatorial waves. As an example Mixed-Rossby-Gravity (MRG) waves are

- arranged antisymmetrically to the equator, whereas Kelvin waves have a symmetric appearance. Therefore, this method decomposes the atmospheric field into a symmetric and an antisymmetric portion and the corresponding power spectra are also separated into symmetric and antisymmetric parts. In their analysis they also found that several statistically significant spectral peaks in the wavenumber-frequency spectra are clustered along the dispersion curves of the equatorially trapped
- waves. Following the simplified equatorial wave theory (e.g. Matsuno, 1966; Lindzen and Holton, 1968) the equatorial wave modes are based on the equatorially trapped solutions of the shallow water equations, which are determined by four parameters: the meridional mode number n, the frequency  $\omega$ , the planetary zonal wavenumber k, and the "equivalent depth" of the layer of the "shallow" fluid. In our study we consider three different equivalent depths of 10, 50 and 250 m. For the sym-
- metric power spectra we utilise n = 1 corresponding to Kelvin waves, and for the antisymmetric case the meridional mode number is n = 0, which corresponds to MRG waves for negative (westward) zonal wavenumbers and to inertio-gravity (IG) waves for positive (eastward) zonal wavenumbers, respectively. For further details on the decomposition and calculation procedure we refer to Wheeler and Kiladis (1999).

### 125 3 Decadal Variability

Figure 1(a) shows the period-latitude distribution of the logarithm of the decadal power spectrum of the zonal wind component at 30 hPa derived from the monthly dataset of ERAI spanning the period from 1990 to 2010. For illustration purposes, the period is given at a logarithmic axis and therefore additionally the power is scaled by the frequency. This is required to preserve area fidelity.

Considering the power spectrum  $P(\omega, k)$  as a linear function of frequency  $\omega$  and wavenumber k, the power between two frequencies  $\omega_1$  and  $\omega_2$  is then given as

$$\int_{\omega_1}^{\omega_2} P(\omega, k) d\omega. \tag{1}$$

As a concrete example, this is equivalent to an area enclosed by the black curve given in Fig. 2. If a logarithmic frequency scale is now applied the power has to be displayed as  $\omega P(\omega, k)$  in order to preserve the variance, which is regarded as equal to the enclosed area. This becomes clear by the equation

$$P(\omega, k)d\omega = \omega P(\omega, k)d(ln\omega).$$
<sup>(2)</sup>

Clear features can be detected in the decadal power spectrum (Fig. 1(a)). For example the annual cycle and semi annual cycle clearly emerge with large peaks at the 12 months and 6 months pe-

- riod, respectively exceeding values of  $1 \log_{10} \frac{m^2}{s^2}$ . Large values in the power spectrum here means that a large proportion of the entire variability of a decade is located at these periods. Furthermore, relatively large variability up to approximately  $-0.8\log_{10} \frac{m^2}{s^2}$  can be found in the polar to mid latitude regions, in particular at periods smaller than 12 months. The detected large variability here is reasonable, since these regions are associated with the appearance of planetary waves, which can
- produce a corresponding response in the strength and temperature of the polar vortex contribution to the variability (e.g.Widnall and Sullivan, 1973; Schoeberl and Hartmann, 1991). In general, comparing both hemispheres at higher latitudes, the northern hemispheric stratosphere bears on average a higher variability than the southern hemisphere. This is reasonable since it is known that the Northern polar vortex is more variable compared to its Southern hemispheric counterpart (Holton et al., 2003).

Apart from high latitudes, a striking variability pattern is situated in the tropics around the 24 months period. By means of a multi-linear-regression model we could prove that this tropical feature is mainly attributable to effects of the QBO (not shown). Additionally, statistical considerations provive that this QBO signal can clearly be distinguished from background noise. The background

noise is simply derived by smoothing the power spectra many times with a 1-2-1 filter. In our case we use 40 passes in order to eliminate any deterministic signal from the raw spectra. In Fig. 2 it can clearly be seen that the raw spectrum in the tropical latitudes significantly exceeds the level of the background noise spectrum at periods around 24 months.

The decadal power spectra derived from model simulations (Fig. 1b - c) give reasonable results compared to the reanalysis dataset. However, a closer inspection shows that in particular the LR simulation, the model configuration with a lower vertical resolution, is not capable to capture the QBO signal (see Fig. 1(c)). Moreover, in all model configurations the intra-annual variability (periods 

in the northern hemisphere. Differences appear in particular in the southern hemispheric at higher latitudes, where the models overestimate intra-annual and intra-seasonal variability compared to the reanalysis data, indicating model deficits in the representation of the southern hemispheric polar vortex.

#### 4 Tropical Variability

#### 170 4.1 Raw and Background Spectra

One of the most prominent phenomenon of tropical stratospheric dynamics is the quasi-biennial oscillation (QBO). The QBO comes close to exhibiting periodic behaviour in the mean zonal winds of the equatorial stratosphere. Zonally symmetric about the equator easterly and westerly wind regimes alternate regularly with periods varying from about 22 to 34 months. It is demonstrated

- that no plausible zonally symmetric advection process could explain this interannual variability, but there must be a vertical transfer of momentum by eddies, which produce this oscillation (e.g. Holton and Hakim, 2013). Observational and theoretical studies have confirmed that vertically propagating equatorial Kelvin and MRG waves provide a significant fraction of the zonal momentum sources necessary to drive the QBO (Baldwin et al., 2001; Kim and Chun, 2015). Kelvin waves pro-
- vide eastward momentum whereas MRG waves provide momentum for the westward acceleration. Kelvin and MRG waves were observed in particular on the intra-seasonal timescale (Andrews et al., 1987).

In the previous section, it has been shown that for the decadal timescale in the reanalysis data the variability at around the 24 months period has a considerable contribution to the tropical stratospheric

power spectrum, which mainly can be attributed to effects of the QBO. However, in the LR model configuration no QBO signal occurs in the corresponding power spectra.

To examine possible reasons for this, we use the wavenumber-frequency power-spectra analysis method of Wheeler and Kiladis (1999). This method was introduced for tropospheric levels, but now is applied to a stratospheric level. It is shown that this the wavenumber-frequency power-spectra anal-

- ysis is capable to analyse the relevant waves of the QBO, in particular the Kelvin and MRG waves. Figure 3 shows contours of the logarithm of the power in the antisymmetric, the symmetric components and the background of tropical (15°N to 15°S) temperature at 30 hPa height, calculated for a 6 hourly dataset of ERAI spanning from 2000 to 2009. The background spectrum is derived from averaging the antisymmetric and symmetric spectra, followed by a smoothing procedure analogously
- to the afore described method for the decadal power spectra. Inspection of the background spectrum (Fig. 3(c)) reveals a smooth and basically "red" noise character, with a slight eastward shift. Regarding the raw power spectra, clear differences in their pattern arise. In the symmetric component (Fig. 3(a)) most of the power is situated at small zonal wavenumbers (< 5) and larger periods (6 to 30 days), which are characteristic scales for planetary and synoptic waves. A clear domination of</p>

- eastward propagating waves can be seen. Their power is accumulated between the dispersion curves of Kelvin waves with equivalent heights of 50 m to 250 m and the amplitudes are decreasing with increasing wavenumber and frequency. But also westward travelling waves can be detected, which are mainly located at planetary wave scales with wavenumbers from -5 to -1 and low frequencies up to 0.1 cycles per day (cpd). The antisymmetric component is given in Fig. 3(b). In contrast to
- the symmetric component the majority of the power is accumulated within the westward propagating waves and their distribution is well captured by the dispersion curves of MRG waves. Apart from this, a local maximum is situated at planetary scale waves with periods of approximately one month (corresponding to frequencies of about 0.1 cpd) and smaller zonal wavenumbers (wavenumbers 1-3). Nonetheless, most of the variability can predominantly be found within the dispersion
- curves of MRG waves with equivalent depths of 250 m to 50 m and periods between 3 and 6 days (corresponding to 0.2 0.4 cpd).

# 4.2 Significant power spectra

For statistical considerations and for a clearer representation of equatorial waves, only the relative power is discussed in the following. Relative power is derived from the raw power spectrum which

- is normalised at each wavenumber and frequency by the corresponding background power. The theoretical absolute number of dof is 2 (space and time) x 10 (latitudes) x 10 (years) x 365/96 (time segments length) ≈ 760. Regarding the corresponding symmetric and antisymmetric spectra and considering that due to averaging the latitudes are not independent from each other, a very conservative estimate for the respective dof would be 76 (2 (space and time) x 1 (latitude) x 10 (years) x 365/96 (time segments length)). To determine the threshold for a certain confidence level
- we follow Warner (1998) and Bloomfield (2004):

$$\frac{S}{\tilde{S}} \le \frac{dof}{\chi_{\alpha}^2},\tag{3}$$

where S is the raw spectra, S̃ is the background spectra and χ<sup>2</sup><sub>α</sub> is the confidence level of the chi-squared distribution. Thus, under the assumption of 76 dof a relative power of approximately 1.3 can
be related to a 95% significance level. This implies values in the ratio of raw power to background power (Figs. 4 and 5) exceeding 1.3 are statistically distinguishable from the background.

The relative spectrum of the symmetric component of ERAI (Fig.4(a)) reveals that the majority of significant peaks arise within the dispersion curves of Kelvin waves with equivalent depths from 250 to 50 m. Inspection of the corresponding power spectra of the model simulations (Fig. 4(b) and Fig. 4(c)) yields a qualitatively good agreement with the reanalysis data, with respect to the pattern

and shape of the significant power spectral peaks. However, the relative power which is attributed to Kelvin waves with frequencies larger than 0.2 cpd tends to be underestimated. In the LR model this underestimation is even more pronounced.

- The statistical significant antisymmetric power spectrum of ERAI is shown in Fig. 5(a). It can be clearly seen that most of the significant variability is located within the dispersion curves of MRG and IG waves with equivalent depths of 250 and 50 m. The antisymmetric component of the relative power in the BASE simulation (Fig. 5(b)) compares fairly well to the reanalysis data. Analogous to the relative power spectrum of ERAI, most of the significant signals are found within the dispersion curves of the MRG and IG waves with 250 and 50 m. However, in the model the distribution of
- significant power spectral peaks is more "compact" meaning the bandwith of frequency as well as wavelengths covered by significant patterns, is more narrow compared to ERAI. In the LR simulation this "compactness" is even more pronounced (Fig. 5(c)). Here, the significant peaks of the power spectrum are predominately situated within the westward components with periods between 6 and 30 days and wavenumbers up to -5. Furthermore, the relative power in the LR configuration is
- comparatively small, meaning the values of the raw spectrum are closer to the background spectrum, which is distinctly different to the results derived from the reanalysis data.

As demonstrated, the LR model with a coarser vertical resolution is not capable to reproduce the low-frequency, tropical variability signal, which is attributed to the QBO. In particular the underestimation of the antisymmetric component of the power spectra is very likely to account for model

- deficits in the LR configuration. By using numerical sensitivity experiments, Giorgetta et al. (2002, 2006) pointed out that a vertical resolution of at least 1 km is necessary in order to simulate a QBO signal in a general circulation model. Further Boville and Randel (1992) showed for their numerical model, that a vertical grid spacing of 1 km or less is required to properly simulate the divergence of the momentum flux of equatorial waves, which is believed to be important to force the QBO.
- Recalling the vertical resolution of approximately 2 km at stratospheric levels in the LR model configuration, the inability to internally generate a QBO signal is in line with the findings of the afore mentioned studies.

Pioneering work in the field of linear wave theory of equatorial waves was contributed by e.g.
Matsuno (1966) and Lindzen and Holton (1968). They present theoretical solutions for equatorially trapped waves based on the linear shallow water equations on an equatorial β plane. After Andrews et al. (1987) the vertical wavenumber m of Kelvin waves is given as

 $m = \frac{2\pi}{\lambda_z} = \left(\frac{N^2}{gh} - \frac{1}{4H_s^2}\right)^{1/2},$ (4)

whereas for MRG waves, in particular for the modes with westward phase speed with respect to the mean flow, the vertical wavenumber m is

$$m = \frac{2\pi}{\lambda_z} = N \left[ \omega - k\overline{u} \right]^{-2} \left\{ \beta + \left[ \omega - k\overline{u} \right] k \right\}.$$
(5)

Here  $\lambda_z$  is the vertical wavelength,  $N^2 = -g/H_s * \frac{d \ln \theta}{d \ln p}$  is the buoyancy frequency squared, g is the acceleration due to gravity, h the equivalent depth,  $H_s$  is the scale height,  $\theta$  is potential temperature,  $\beta$  is the beta coefficient,  $\overline{u}$  is the zonal wind, k and  $\omega$  are the zonal wavenumber and

- the corresponding angular frequency, respectively. Fig. 6 shows a 10 year climatological mean 270 of the vertical distribution of  $\lambda_z$  for Kelvin and MRG waves derived from Eq. 4 and 5 utilising ERAI. Only equivalent depths between 50 and 250 m were considered, since within these values the majority of the statistically relevant waves were detected in the power spectra shown above (see Figs. 4(a), 5(a)). Further, the calculation of  $\lambda_z$  for MRG waves only considers the most relevant zonal wavenumbers (k = [-6; -1]). In general, the vertical wavelengths of Kelvin waves pre-
- sented here are larger than the vertical wavelengths of MRG waves. This is in accordance with the findings of Boville and Randel (1992), who showed a similar behaviour of equatorial trapped waves in their model simulation. At lower stratospheric levels (e.g. 50 hPa) the calculated  $\lambda_z$  of ERAI corresponds fairly well to  $\lambda_z$  identified in observational data from the equatorial stratosphere (e.g. Yanai and Maruyama, 1966; Wallace and Kousky, 1968). The observed vertical wavelengths of
- lower stratospheric Kelvin waves are about 6 to 10 km and corresponding MRG waves are about 4 to 8 km (Andrews et al., 1987). Inspection of the vertical wave lengths calculated from ERAI at higher altitudes reveals that  $\lambda_z$  of Kelvin waves keep a more or less constant level of about 6 to 14 km. However, in the case of the MRG waves a clear decrease of  $\lambda_z$  at lower to mid stratospheric levels is identified. In particular at about 30 hPa the largest part of the vertical wavelengths are smaller than
- 4 km. Furthermore, analysis of  $\lambda_z$  of MRG waves with an equivalent depth of 50 m yields vertical wavelengths of approximately 2 km at stratospheric altitudes.

So, by means of the wavenumber-frequency power-spectra analysis introduced by Wheeler and Kiladis (1999), which was now applied for a stratospheric level, we could demonstrate that for ERAI the relevant MRG waves have equivalent depths between 50 m to 250 m. The calculations above showed that a vertical grid spacing of 1 km or less is necessary in order to resolve these waves.

#### 5 Conclusion and Discussion

The decadal power spectral analysis presented here is a useful method to gain a comprehensive overview of stratospheric variability on the decadal timescale and to identify the corresponding relevant stratospheric processes. Applying this decadal power spectral analysis to the ERAI zonal mean wind field at 30 hPa reveals that a large fraction of the variability is situated at the 12 and 6 months

period. Further large variability can be found in the extra tropical region at intra-annual timescales and in particular in the tropics at the 24 months period, which is mainly attributable to the QBO.

Regarding decadal variability of model simulations, clear differences between the BASE (the configuration with a relatively high vertical resolution in the stratosphere) and the LR (the configuration

with a coarser vertical resolution) simulation appear. The BASE simulation qualitatively captures the characteristics of stratospheric variability, whereas the LR simulation cannot adequately represent the tropical variability and the QBO signature in the decadal power spectrum is missing.

In order to analyse the drivers of the QBO, the seasonal timescale needs to be considered. Therefore, the frequency-wave spectra analysis of Wheeler and Kiladis (1999) is applied for stratospheric

- levels, to examine the variance of Kelvin and MRG waves. Calculations based on linear wave theory reveal that for the ERAI reanalysis data the vertical wavelengths of the relevant MRG waves range between 2 and 4 km. This corresponds well to the scales found by observations (Andrews et al., 1987). In order to capture the characteristics of these waves and its corresponding stratospheric equatorial variability, models need to provide a vertical resolution of at least 1 km.
- With respect to the LR simulation, the underestimated variance of MRG waves can be explained 310 by the fact that the vertical resolution in the LR model configuration is insufficient in order to resolve the necessary wave spectra. Therefore, in the LR simulation a substantial proportion of momentum is missing and no QBO can internally evolve. However, other points could affect this model deficit. For example Holt et al. (2016) showed that the divergence of the momentum flux of equatorial waves
- considerably depends on the vertical model resolution. Boville and Randel (1992) demonstrated that it is the momentum flux of the MRG waves, which is influenced by the vertical spacing of model levels. So further analyses are needed to quantify these aspects in more detail.

Apart from large scale Kelvin and MRG waves (which are the main drivers) also momentum of inertio-gravity (IG) waves is required in order to obtain a realistic QBO (e.g. Dunkerton, 1997;

Sato and Dunkerton, 1997). Indeed, the IG waves are also detectable in the tropical power spectrum of the reanalysis data shown above, i.e. significant power of the symmetric component (see Fig. 4(a) values  $\geq$  1.3) located at westward zonal wavenumbers with high frequencies between 0.55  $\sim$ 0.75 cpd are attributable to IG waves (Sato and Dunkerton, 1997; Wheeler and Kiladis, 1999). In our model simulations this feature of IG waves is less pronounced. It is known that this type of IG

waves are strongly coupled to convection (Holton and Hakim, 2013; Wheeler and Kiladis, 1999). Convection is parameterised in our simulations and therefore relevant processes for the IG waves could be not so well represented, which would lead to this underestimation in the power spectrum. This underestimated variability of IG waves also could explain the differences in strength and shape of the QBO signal detected in the decadal power spectrum between the BASE (Fig. 1(b)) and ERAI

(Fig.1(a)).

To conclude, this work shows that in the context of decadal predictions the QBO has an important role. For the intra decadal timescale the QBO dominates the tropical stratospheric variability. In concert with other studies we could estimate a threshold of 1 km or less for vertical grid spacing, which is required for models to generate a QBO (e.g. Boville and Randel, 1992; Giorgetta et al.,

2002, 2006). Differently to these studies our estimation is based on calculations derived from power spectral analysis and linear wave theory. Nonetheless, further efforts are needed, in particular regarding gravity waves and their corresponding parameterisations, in order to robustly reproduce the tropical stratospheric variability.