# Peer review of "Stratospheric Variability at a glance - Analysis of the intra decadal timescale and the QBO"

_Atmospheric Chemistry and Physics, 2016_

## Referee Comment (RC1) · Anonymous Referee #1 · 23 Nov 2016

This paper analyzes an intradecadal spectrum of the stratosphere using a reanalysis and compares it with model simulations. One of the main differences in the intradecadal variability between the reanalysis and model results is associated with the QBO in the tropics. The authors assessed simulated equatorial waves in intraseasonal spectra of the middle stratosphere, and attributed the defect in simulating the QBO to underrepresentation of the equatorial waves in the models.

Major comments :

The above findings, however, have been well known by previous studies, as the authors also mentioned in the paper. The authors did not extend their analysis further, and thus the paper does not seem to provide concrete ideas that can support or add on

our understanding on this topic. Based on the Introduction section, this paper seems motivated by the potential impacts of model-representation of the stratospheric variability upon the (intra) decadal predictability. I agree that this is very interesting topic. However, it might be not easy to address/assess the potential impacts of the stratospheric variability using this relatively short-term (20 years) climate simulation. In addition, by estimating the typical vertical scales of the equatorial waves presented in the reanalysis, the authors discussed the model vertical resolution that is required to simulate the QBO (< 1 km). Regarding this, readers may expect more contents than those presented here (e.g., actual spatiotemporal structures of waves and their propagation in simulations using different stratospheric resolutions) because many recent papers have demonstrated this point even in more details (e.g., Krismer and Giorgetta, 2014; Richter et al., 2014; Anstey et al., 2016). Considering the contents of the paper, I would not recommend to publish this manuscript in ACP.

Specific comments :

The manuscript requires linguistic advice in terms of sentence structures, use of comma, etc.

L16: "are" → "is"

L21: This sentence is not relevant to this paragraph. Please move it to L43.

L26: "showed that"

L41: "QBO and in" → "QBO. In"

L42: Scaife et al. (2014) showed that the skillful prediction of the QBO solely does not guarantee the predictability of the winter surface.

L48: "final section 5" → "final section"

L58: What does the "low-level" mean ? Vertical level ? Or, level of components ?

Please clarify it.

L65: Please provide the full name for IPCC.

L66-67: Please delete this sentence.

L69: Please insert the comma after "spin-up".

L73: Is this the 'effective' resolution? What is the vertical grid spacing here?

L81: Please place this reference to L53.

L86: "squares" → "square"

L95: Delete "and a 6 months".

L100: Please provide the full name for FFT.

L100-101: This is an ordinary way to get the variances. Please delete this sentence.

L102: If you would sum over all wavelengths, you did not have to perform the 2-dimensional FFT for the decadal time series.

L115 and L259: I suspect that the paper by Lindzen and Holton (1968) might be not a proper reference here. I guess that you intended to refer to Lindzen and Matsuno (1968). Please confirm these.

L120: "n = 1" → "n = −1"

L121: "number is" → "number we consider is"

L130-137: This description does not seem really necessary.

L140 and Fig. 1 caption: The log of any variables cannot have a unit (unless you specifically define the quantity with a reference power, like decibel [dB]).

L140-141: So obvious statement. Please remove it.

L142: "variability" → "power"

L143: Delete "in particular".

L145: Please rephrase "polar vortex contribution to the variability" to clarify the meaning.

L147: Include "for periods shorter than 6 months" after "latitudes"

L154: "provide"

L153-158: This fact is very well known. The authors could just add some references here rather than describe this in detail. In addition, Fig. 2 does not address more than what we know.

L165: "... hemisphere high latitudes"

L166: "intra-annual and intra-seasonal": What are the difference?

L172: "The QBO comes close to exhibiting ...": Please make the sentence clear.

L173: Delete "about the equator".

L188-189: Please move this sentence to Section 2.

L189: Delete "this".

L199: Synoptic waves may have zonal wavenumber larger than 4 (k = 4 corresponds to the wavelength of 10000 km in the tropics). Please delete "and synoptic".

L208: "0.1" → "0.03"

L230: Delete "pattern and" unless the "pattern" here has a different meaning from the "shape" and you will describe both.

L239-241: I could not understand this sentence. Please rewrite it.

L244: "30" → "3" (or 2 days?)

L258: "linear equatorial wave theory was"

[Figure]

L266: Note that the equation for $N^2$ written here is correct only if H is defined as RT/g, where T is a mean temperature that actually represents the temperature in the tropics at this altitude (Eq. 1.1.13 on p6 in Andrews et al., 1987).

L269: "Figure 6"

L269: How did you obtain the climatological mean of the vertical wavelengths of MRG waves? It is spectral-power-weighted mean?

L280: "corresponding" → "those of"

L285-286: Please rephrase the sentence.

L289: "to" → "and"

L300: "simulations appear."

L304: "frequency-wavenumber"

L318: Please delete "(which are the main drivers)" because it is believed that the IG and mesoscale gravity waves may also provide large momentum, as well as the Kelvin waves.

L331: I do not agree that this paper shows it.

L336-338: This is a rather unexpected statement because there was no discussion regarding the gravity wave parameterization before this.

References :

Andrews, D. G., Holton, J. R., and Leovy, C. B.: Middle atmosphere dynamics, 489 pp., Academic, San Diego, Calif, 1987.

Anstey, J. A., Scinocca, J. F., and Keller, M.: Simulating the QBO in an atmospheric general circulation model: Sensitivity to resolved and parameterized forcing, J. Atmos. Sci., 73, 1649-1665, 2016.

Krismer, T. R. and Giorgetta, M. A.: Wave forcing of the quasi-biennial oscillation in the Max Planck Institute Earth System Model, J. Atmos. Sci., 71, 1985-2006, 2014.

Richter, J. H., Solomon, A., and Bacmeister, J. T.: On the simulation of the quasi-biennial oscillation in the Community Atmosphere Model, version 5, J. Geophys. Res. Atmos., 119, 3045-3062, 2014.

---

## Author Comment (AC1) · 9 Dec 2016

This paper analyzes an intradecadal spectrum of the stratosphere using a reanalysis and compares it with model simulations. One of the main differences in the intradecadal
variability between the reanalysis and model results is associated with the QBO in the tropics. The authors assessed simulated equatorial waves in intraseasonal spectra of the middle stratosphere, and attributed the defect in simulating the QBO to underrepresentation
of the equatorial waves in the models.

*We want to thank referee #1 for the valuable comments and the detailed review.*
*We have to admit, that we missed referring our work to some of the recent, relevant studies (i.e. Krismer and Giorgetta 2014). In a revised version we will change this accordingly. In particular our conclusion would benefit from considering their results.*

Major comments :
The above findings, however, have been well known by previous studies, as the authors also mentioned in the paper. The authors did not extend their analysis further, and thus the paper does not seem to provide concrete ideas that can support or add on our understanding on this topic.

*Indeed, the finding of a threshold for a vertical resolution is well known. However, the discussion of underlying reasons and mechanisms is still highly topical, which you also indicated by your given references. For example gravity wave parameterization with respect to vertical resolution (Jadwiga et al. (2014) and Anstey et al. 2016) or the very detailed study of Krismer and Giorgetta (2014) about the analysis of resolved wave (in particular Kelvin wave) forcing of the QBO in their model system .*
*Furthermore, to our knowledge this is the first time that the relevance of the QBO has been addressed by power spectral analysis for the intra decadal time scale.*

Based on the Introduction section, this paper seems motivated by the potential impacts of model-representation of the stratospheric variability
upon the (intra) decadal predictability. I agree that this is very interesting topic. However, it might be not easy to address/assess the potential impacts of the stratospheric variability using this relatively short-term (20 years) climate simulation.

*Fair comment! The introduction could suggest that this study has a specific focus on the decadal prediction. However, the decdal time scale is our main motivation. We demonstrate that on the decadal time scale, globally the QBO plays an important role. This is shown by our decadal power spectral analysis, where a great fraction of the power can be find near the 24 months period in the tropics. In a revised version we will certainly rephrase our introduction.*

*Further, in our opinion a 20 years climate simulation are a good time range to investigate processes of the intra decadal (<10 years). In particular we could demonstrated that our signals derived from the decadal power spectral analysis are robust and statistically significant.*

In addition,
by estimating the typical vertical scales of the equatorial waves presented in the reanalysis, the authors discussed the model vertical resolution that is required to simulate

the QBO (< 1 km).
Regarding this, readers may expect more contents than those
presented here (e.g., actual spatiotemporal structures of waves and their propagation
in simulations using different stratospheric resolutions) because many recent papers
have demonstrated this point even in more details (e.g., Krismer and Giorgetta, 2014;
Richter et al., 2014; Anstey et al., 2016).

*We agree, that the reader could, as always, expect more content in particular aspects. However, in our opinion this study is self-contained. May we briefly explain this. First of all, we started with the analysis of the intra decadal power spectrum. We show differences between reanalysis data and our model simulation with different vertical resolutions. The QBO signal only could be reproduced by the model with high vertical resolution. To analyse the causes we applied the Wheeler and Kiladis (1999) method. We compared the wave spectra of our two model simualtions to reanalysis data. We can point out that a low vertical resolution of the model lacks of representing the anti-symmetric wave spectra, in particular the power of the MRG waves were under represented. The analysis of anti-symmetric wave spectra of the reanalysis data narrows the range of the equivalent depth relevant for the QBO. This narrowed range of equivalent depth we use as input for our calculation for the vertical wavelengths of MRG waves. With this calculation we can show that the relevant wave spectra of MRG waves needs at least to resolve waves with a vertical wave length of 2 km, which means that the vertical model discretisation need to be less than 1 km. These aspects of MRG waves, the approach which lead to this finding is new and so far not documented.*

*So in simple words, we came from the intra decadal time scale, detected the relevance of the QBO and found model problems representing the QBO. Compared to reanalysis majorly the MRG waves are underestimated. We could pin point this problem to insufficient vertical model resolution. Therefore, we think this study is self-contained.*

Considering the contents of the paper, I would not recommend to publish this
manuscript in ACP.

*We do not agree with this conclusion. As written above, we demonstrated new aspects in analysing the QBO. In particular the relevance of vertical resolution for MRG waves are pointed out. Further, the relevance of the QBO for the intra decadal timescale is shown. Nonetheless, for a revised version we must refer in more detail to the references, provided by the reviewer #1*

L100-101: This is an ordinary way to get the variances. Please delete this sentence.
*We did this for clarification. We think this is necessary, because we also found different definitions with respect to term variance.*

L102: If you would sum over all wavelengths, you did not have to perform the 2-dimensional FFT for the decadal time series.
*The 2D FFT for this approach would be unnecessary. However, our code use 2D FFT since, for other application 2D FFT is necessary, in oder to analyse characteristic bandwidths. So this is only the correct description of the process of our calculation.*

L115 and L259: I suspect that the paper by Lindzen and Holton (1968) might be not a proper reference here. I guess that you intended to refer to Lindzen and Matsuno (1968). Please confirm these.
*will be revised*

L120: "n = 1" ! "n = -1"
*will be revised*

L121: "number is" ! "number we consider is"
*will be revised*

L130-137: This description does not seem really necessary.
L140 and Fig. 1 caption: The log of any variables cannot have a unit (unless you specifically define the quantity with a reference power, like decibel [dB]).
*will be revised*

L140-141: So obvious statement. Please remove it.
*will be revised*

L142: "variability" ! "power"
*will be revised*

L143: Delete "in particular".
*will be revised*

L145: Please rephrase "polar vortex contribution to the variability" to clarify the meaning.
*will be revised*

L147: Include "for periods shorter than 6 months" after "latitudes"
*will be revised*

L154: "provide"
*will be revised*

L153-158: This fact is very well known. The authors could just add some references here rather than describe this in detail. In addition, Fig. 2 does not address more than what we know.
*Fig. 2 does provide new aspects! It is shown that the spectra around the 24 months period is one of the mayor contributions of the intra decadal time scale and clearly can be distinguished from back ground noise.*

L165: "... hemisphere high latitudes"
*will be revised*

L166: "intra-annual and intra-seasonal": What are the difference?
*>12 months; >3 months*

L172: "The QBO comes close to exhibiting ...": Please make the sentence clear.
*will be revised*

L173: Delete "about the equator".
*will be revised*

L188-189: Please move this sentence to Section 2.
*will be revised*

L189: Delete "this".
*will be revised*

L199: Synoptic waves may have zonal wavenumber larger than 4 (k = 4 corresponds to the wavelength of 10000 km in the tropics). Please delete "and synoptic".
*will be revised*

L208: "0.1" ! "0.03"

*will be revised*

L230: Delete "pattern and" unless the "pattern" here has a different meaning from the "shape" and you will describe both.
*will be revised*

L239-241: I could not understand this sentence. Please rewrite it.
*will be revised*

L244: "30" ! "3" (or 2 days?)
*will be revised*

L258: "linear equatorial wave theory was"
*will be revised*

L266: Note that the equation for $N_2$ written here is correct only if H is defined as RT/g, where T is a mean temperature that actually represents the temperature in the tropics at this altitude (Eq. 1.1.13 on p6 in Andrews et al., 1987).
*We use this only for a crude estimation.*

L269: "Figure 6"
*will be revised*

L269: How did you obtain the climatological mean of the vertical wavelengths of MRG waves? It is spectral-power-weighted mean?
*Please read section 2 from line 265 on.*

L280: "corresponding" ! "those of"
*will be revised*

L285-286: Please rephrase the sentence.
*will be revised*

L289: "to" ! "and"
*will be revised*

L300: "simulations appear."
*will be revised*

L304: "frequency-wavenumber"
*will be revised*

L318: Please delete "(which are the main drivers)" because it is believed that the IG and mesoscale gravity waves may also provide large momentum, as well as the Kelvin waves.
*We totally agree with the referee with respect to a general statement regarding the QBO. However, in our analysis we refer to lower stratospheric altitudes. IG and mesocale g-waves interacts in general in higher altitudes. We will revised this accordingly.*

L331: I do not agree that this paper shows it.
*Will be revised. The QBO is important for the intra decadal time scale and therefore will have great potential for the decadal prediction.*

L336-338: This is a rather unexpected statement because there was no discussion regarding the gravity wave parameterization before this.
*Will be revised and further information will be included.*

---

## Referee Comment (RC2) · Anonymous Referee #2 · 17 Dec 2016

The authors present several good quality plots that would be of interest as a kick-off to a model intercomparison activity. However, the basic approach of using Hyashi spectrum / Wheeler and Kiladis style analysis has been widely applied to different fields from surface precipitation to top-of-atmosphere OLR as well as vertically varying levels. I myself considered it a sufficiently well-trodden route to suggest that a 2010 summer placement student follow published work of Ern et al. to investigate the propagation of Kelvin waves in two different resolution versions of our GCM and start with spectral plots such as these at varying heights in the model stratosphere, which suggests to me that such use is fairly commonplace.

Refs: M. Ern et al., Atmos. Chem. Phys., 8, 845–869, 2008 M. Ern and P. Preusse, Atmos. Chem. Phys., 9, 3957–3986, 2009

[Figure]

I think the authors would have to concede that either of these papers is considerably more substantial than the manuscript submitted here.

If the aim is to keep the paper short (and as a reviewer I have no objection to that) it really does need to have more impactful (well-supported and focused) conclusions than the authors presently offer. Starting an introduction with the statement that the QBO is a dominant source of internal variability in the tropical lower stratosphere would be fair enough: presenting it as a conclusion appears less than insightful.

Likewise, the very basic statements that QBOs depend on waves on a range of scales and that vertical resolution will impact the representation in models were made around the time of the 2001 Baldwin et al. review and can hardly be seen as 'novel' today.

The precise statement that your manuscript supports would be that "Given the horizontal resolution of our EMAC configuration a vertical resolution longer than about 2km is not sufficient to sustain a QBO at 30hPa whereas a resolution about 1km appears to be." This is a much more qualified and model specific statement than you make on several occasions but actually it could be a good starting point for extending the work in one of a number of directions, such as investigating other levels in your model to distinguish issues of wave generation versus propagation, mean flow structure, looking at interactions between resolved waves and any subgrid parametrizations (you say nothing about these), further variation of resolution to refine the transition and even making comparisons against other models.

In conclusion, I agree with the authors that there is interest in the general topic of modelling the QBO and understanding the processes by which models arrive at their diverse representations but I would encourage them to ask what is the key message with which they would like to really catch a reader's attention in this manuscript, whether they choose to tighten or expand it, and what information about or from the experiment really adds to that core message that they want people to buy in to.

---

## Referee Comment (RC3) · Anonymous Referee #3 · 22 Dec 2016

The purpose of this paper is unclear to me. Comparing two versions of an atmospheric general circulation model (AGCM), one with high vertical resolution and one with coarse vertical resolution (and no other difference between the model versions), it is shown that power at the QBO frequency is absent in the model with coarse vertical resolution. The importance of vertical resolution for simulating the QBO is already a well known result in the QBO literature. It is then shown that the coarse model shows weaker mixed-Rossby-Gravity wave (MRGW) activity than the fine-resolution model and ERA-Interim. This is expected from linear wave theory, and is shown in this paper but it is also well known from the literature (including studies cited here). It is then surmised that inadequate representation of the MRGW is responsible for the lack of a QBO in the coarse model. No calculations of the zonal momentum forcing due to resolved waves are presented, so it is left unclear whether the forcing from MRGW is

important relative to other wave types. Gravity waves are not discussed at all, except to be given cursory mention in the final paragraph of the conclusions, and it's not stated whether the model uses parameterized non-orographic gravity wave drag although I imagine that since the horizontal resolution is relatively coarse (T42) it probably does.

The quality of writing is generally good, and the figures are clearly presented. The authors cite most of the relevant literature. However, since the results do not appear to add anything new to the literature on the QBO, I must recommend rejection of this paper.

Here are a few other comments by line number:

90-106: The power spectral method is not clearly described. It's not explained why the calculation is done for 10-year segments in continuously shifting windows. At line 105, it's not clear what "indepedent realizations" means in this context.

153-154: Why is a statistical significance test used (for ERA-Interim, Fig 2)? It's reasonably well established that the QBO exists.

178: How big is a "significant fraction" of the zonal momentum forcing? Why is there no mention of gravity waves?

331: "To conclude, this work shows that in the context of decadal predictions the QBO has an important role." Decadal predictions of what? If the troposphere, this has not been shown.

---

## Author Comment (AC2) · 22 Dec 2016

Reply Anonymous Referee #2:
We want to thank Anonymous Referee #2 for the comments. And we appreciate your suggestions. Indeed, the finding of a vertical threshold for the vertical resolution of GCMs are well known. However, to our opinion our manuscript presents different aspects and novel approach to arrive to this well known conclusion. May we briefly summarize these aspects and explain our approach.

We started with the analysis of the intra decadal power spectrum and could show the domination of the QBO in the spectrum. To our knowledge this is the first time that the relevance of the QBO has been addressed by power spectral analysis for the intra decadal time scale. The decadal time scale is our motivation, and therefore the title and introduction could mislead the reader. So in a revised version we would make this more clear.

The missing QBO signal in the lower resolution model for the intra decadal power spectrum is expected due to the fact that the vertical model resolution is coarser than 2 km. Also the followed analyses using the method of Wheeler and Kiladis (1999) of symmetric and antisymmetric power spectrum is not novel. However, we now can point out that a low vertical resolution of the model lacks of representing the antisymmetric wave spectra, in particular the power of the MRG waves were under represented. From ERAI data we derived the statistically relevant waves in the antisymmetric wave spectrum. Following linear wave theory these relevant waves are characterized by a certain range of equivalent depth. This certain range of equivalent depth we use as input for our calculation for the vertical wavelengths of MRG waves. With this calculation we can show that the wave spectrum of MRG waves derived from ERAI, needs at least to resolve waves with a vertical wave length of 2 km. For alls numerical models in general, this means that the vertical model discretization need to be less than 1 km in order to resolve the relevant wave spectrum of MRGW. These aspects of MRG waves, the approach which lead to this finding is new and so far not documented.

The discussion need to be expanded. In particular regarding subgrid scale parametrization and also possible weaknesses of ERAI data with respect o satellite data as shown in the references mentioned by reviewer 2#. This would be done in a revised version.

For a revised version we need to pin point these new aspects more clearly as it is said by reviewer #2.

---

## Author Comment (AC3) · 23 Dec 2016

Reply Anonymous reviewer 3#

We want to thank Anonymous Referee #3 for the comments. Indeed, the finding of a vertical threshold for the vertical resolution of GCMs are well known. However, to our opinion our manuscript presents different aspects and novel approach to achieve this well known conclusion. May we briefly summarize these aspects and explain our approach.

We started with the analysis of the intra decadal power spectrum and could show the domination of the QBO in the spectrum. To our knowledge this is the first time that the relevance of the QBO has been addressed by power spectral analysis for the intra decadal time scale. The decadal time scale is our motivation, and therefore the title and introduction could mislead the reader. So in a revised version we would make this more clear.

The missing QBO signal in the lower resolution model for the intra decadal power spectrum is expected due to the fact that the vertical model resolution is coarser than 2 km. Also the followed analyses using the method of Wheeler and Kiladis (1999) of symmetric and antisymmetric power spectrum is not novel. However, we now can point out that a low vertical resolution of the model lacks of representing the antisymmetric wave spectra, in particular the power of the MRG waves were under represented. From ERAI data we derived the statistically relevant waves in the antisymmetric wave spectrum. Following linear wave theory these relevant waves are characterized by a certain range of equivalent depth. This certain range of equivalent depth we use as input for our calculation for the vertical wavelengths of MRG waves. With this calculation we can show that the wave spectrum of MRG waves derived from ERAI, needs at least to resolve waves with a vertical wave length of 2 km. For alls numerical models in general, this means that the vertical model discretisation need to be less than 1 km in order to resolve the relevant wave spectrum of MRGW. These aspects of MRG waves, the approach which lead to this finding is new and so far not documented. For a revised version we need to pin point these new aspects more clearly.

Regarding subgrid scale waves, i.e. parametrized gravity waves our discussion sector need to be expanded and will be added in a revised version.

We do not agree with the reviewer criticism not showing new results.

Reply for comments by line number:

*90-106: The power spectral method is not clearly described. It's not explained why the calculation is done for 10-year segments in continuously shifting windows. At line 105, it's not clear what "indepedent realizations" means in this context.*
We meant statistically independent. Will be revised.

*153-154: Why is a statistical significance test used (for ERA-Interim, Fig 2)? It's reasonably well established that the QBO exists.*
True. However, it is no clear whether the QBO signal is statistically distinguishable from the background noise on a intra decadal time scale. This is an important question and not answered yet.

*178: How big is a "significant fraction" of the zonal momentum forcing? Why is there no mention of gravity waves?*
Aspects parametrization of gravity waves in the model will be added in a revised version.

*331: "To conclude, this work shows that in the context of decadal predictions the QBO has an important role." Decadal predictions of what? If the troposphere, this has not been shown.*
Will be made clearer in a revised version

---

## Author Comment (AC4) · 23 Dec 2016

We want to thank all reviewers for their helpful comments.

It become clear to us that it is hard for the reader to see our main conclusions. However, we cannot agree with the reviewers criticism not presenting new results.

Therefore we want to emphasize the key points, which are to our opinion new and relevant:

the relevance of the QBO on the intra decadal time scale: the QBO signal is significantly distinguishable from background noise, regarding the intra decadal time scale aspects of MRGW and the approach to receive a vertical threshold for numerical models to simulate the QBO: using power spectral analysis method of Wheeler and Kiladis

(1999) to spot the relevant ranges of equatorial waves ; using these identified ranges to calculate the corresponding vertical wave length; receiving a minimum vertical threshold in order to resolve the relevant waves, i.e. relevant power spectrum of MRGW need to include waves with 2 km vertical wave length (this requirement is for the MRGW is also new to us)

So our study does not focus on processes of a specific numerical model. But we can confirm the finding that numerical models need to have at least a vertical resolution of 1 km, based on a 'novel' approach. This 'novel' approach also show new aspects regarding the requirements for the simulation of a QBO.

We hope we can convince you, that is it worth to allow us to revise our manuscript in order to made our new points more clear.

---

## Referee Comment (RC4) · Anonymous Referee #3 · 3 Jan 2017

Thank you to the authors for their quick response to my comments. I'm replying in order to add some further details re. my two main criticisms of the paper.

1) Re. the intradecadal timescale, the Authors replied: "To our knowledge this is the first time that the relevance of the QBO has been addressed by power spectral analysis for the intra decadal time scale."

The QBO's timescale is intradecadal, since it has a period of roughly

2.4 years. The QBO period is variable, and the QBO may be influenced by other low-frequency phenomena (e.g. ENSO) and so the QBO peak will appear somewhat smeared out in a power spectrum. So I do not understand what is novel about Fig. 1 and 2, since it seems to me that the power spectra show exactly what you would expect to see. Power spectra have of course been shown in previous studies (e.g. Pascoe et

al. 2005, Fig 3).

I do not understand why you emphasize "for the intra decadal time scale"

- I presume this means simply that your power spectrum covers periods up to about 10 years. But if there is something special about this approach that I have misunderstood, please explain (if I have missed it, perhaps other readers may also miss it). Finally, re. Fig 2, I don't understand what is the null hypothesis for this statistical test. If the null hypothesis is that there is no QBO, it doesn't seem a very useful test since this existence of the QBO is established beyond any reasonable doubt. The way the paragraph at lines 151-158 is written suggests that your null hypothesis is that there is no QBO. But if this is not what you meant then you need to more clearly describe your null hypothesis.

2) Re. the MRG waves, I agree that you have demonstrated clearly that the representation of these waves in your model differs substantially between the high-vertical resolution and low-vertical resolution model versions. The calculation from linear wave theory (Fig. 6) is useful to explain this result, but it is not in itself new. Boville and Randel 1992, which you cite, already showed this. More importantly, though, your results don't demonstrate that the MRG wave is very important for the QBO in your model or in ERA-Interim. Without determining this, you cannot claim that inadequate resolution of the MRG wave is the key factor determining why vertical resolution is important for modelling the QBO. It has been well known for some time that realistic levels of Kelvin and MRG wave activity are insufficient to drive the QBO by themselves, and substantial forcing from gravity waves is also required.

The forcing by the MRG tends to be smaller than that due to Kelvin waves, e.g. Kim and Chun 2015, Fig 2. To determine how important is the MRG forcing, you need to calculate the terms in the zonal momentum budget. If the MRG turns out to provide a substantial fraction of the wave forcing in your model, then it might be an important factor determining why a QBO occurs at high vertical resolution but not at low vertical

resolution. Note, however, that since changing the vertical resolution of the model also changes the background zonal wind state, in comparing the two model versions you are not only comparing the effects of changed vertical resolution on the waves, but also the effect of a changed background state.